# CURIOSITY-DRIVEN LLM-AS-A-JUDGE FOR PERSONALIZED CREATIVE JUDGMENT

## ABSTRACT

Modern large language models (LLMs) excel at objective tasks such as evaluating mathematical reasoning and factual accuracy, yet they falter when faced with the nuanced, subjective nature of assessing creativity. In this work, we propose a novel curiosity-driven LLM-as-a-judge for evaluating creative writing which is personlized to each individual's creative judgments. We use the Torrance Test of Creative Thinking(TTCW) benchmark introduced in Chakrabarty et al. (2024), which has stories annotated by expert humans across various subjective dimensions like *Originality*, to test our hypothesis. We show that our method enables models across various sizes, to learn the nuanced creative judgments of different individuals, by showing improvements over baseline supervised finetuning(SFT) method across various evaluation metrics like Pearson correlation, Cohen's $\kappa$ and F1 values. Our method is especially useful in subjective evaluations where not all the annotators agree with each other.

## 1 INTRODUCTION

Rigorous, standardized evaluation has repeatedly catalyzed progress in machine learning, from ImageNetRussakovsky et al. (2015) and GLUEWang et al. (2019), driving leaps in the fields of computer vision and Natural Language Processing, respectively. The same effect is evident in objective math reasoning, where benchmarks like GSM8KCobbe et al. (2021), together with RL-trained reasoning models such as OpenAI's o1OpenAI et al. (2024) and DeepSeek-R1DeepSeek-AI et al. (2025) have obtained strong results on hard contests like AIME and IMO.

While robust evaluation metrics exist for objective tasks such as mathematical reasoning and factual verification, subjective tasks like creativity remain difficult to assess reliably. There are many previous works Panickssery et al. (2024a); Wataoka et al. (2025) which show that using Large Language Models(LLM) as a judge prefer their own generations making them unreliable. Despite the success of LLMs on objective benchmarks, they still struggle to evaluate creativity in a manner aligned with human judgment. As shown in Chakrabarty et al. (2024) and Table 12 and Table 2, even state-of-the-art models fall short in consistently evaluating the subjective dimensions of the story as well as a human expert. This can be attributed to the fact that individual preferences shape creativity and rarely align uniformly across people.

To address this gap, we present an enhanced LLM-as-a-judge that not only learns from a diverse pool of annotations but also adapts its scoring to align with individual annotators or experts. This allows for more faithful and preference-aware evaluation of creativity. We emphasize personalization in our framework because the task of assessing subjective criteria is inherently variable across individuals. To this end, we propose a curiosity-driven LLM-as-a-judge for evaluating creativity in text generation, drawing inspiration from the curiosity-based Reinforcement Learning (RL) framework of Pathak et al. (2017). However, unlike the RL setting in Pathak et al. (2017), we reinterpret curiosity as an *belief-shift signal* for creative evaluation. Specifically, when the model is "surprised" by an expert's explanation, it signals a mismatch between the LLM's prior belief and the expert's preference; conversely, low surprise indicates alignment between the LLM and the expert (see Fig 5. To implement this, we first train an Intrinsic Curiosity Model (ICM) that measures the LLM's surprise at a given explanation while simultaneously predicting which expert or annotator produced the explanation. The intuition behind predicting the annotator is that the model can learn which annotator caused the belief shift, allowing it to calibrate the curiosity signal for each annotator

individually, thereby improving personalization. The resulting *curiosity score* is then fed as an auxiliary, self-supervised signal to improve a supervised fine-tuning (SFT) model (see Fig 1).

In our experiments, we establish a baseline using an SFT model that predicts annotators' binary judgments from the story and question (see Fig 3a). To evaluate the effect of curiosity, we enhance this baseline with an ICM-derived curiosity score. More concretely we append the curiosity score to story and question in the baseline model. This helps us do a fair comparison on effect of curiosity signal on the final judgment and thereby measure the lift in performance our methodology provides over the baseline.

We conduct extensive experiments across various model sizes to ensure our method scales well with model size. Since the TTCW dataset size is extremely small, we do a 5-fold cross validation in order to ensure that our results are statistically significant. We also test our method in out-of-distribution scenarios to ensure that our method generalizes well. Averaged across model sizes, ICM significantly improves Pearson correlation and F1 scores. More details about the results can be found in Fig 4.

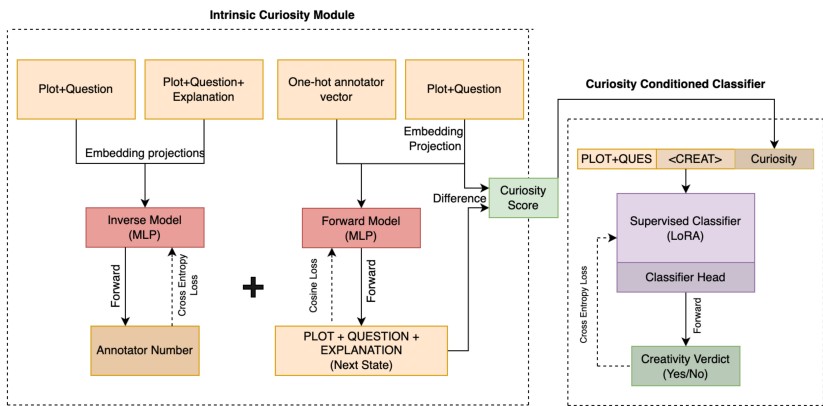

Figure 1: Overview of Architecture during training for Curiosity Driven LLM-as-a-judge

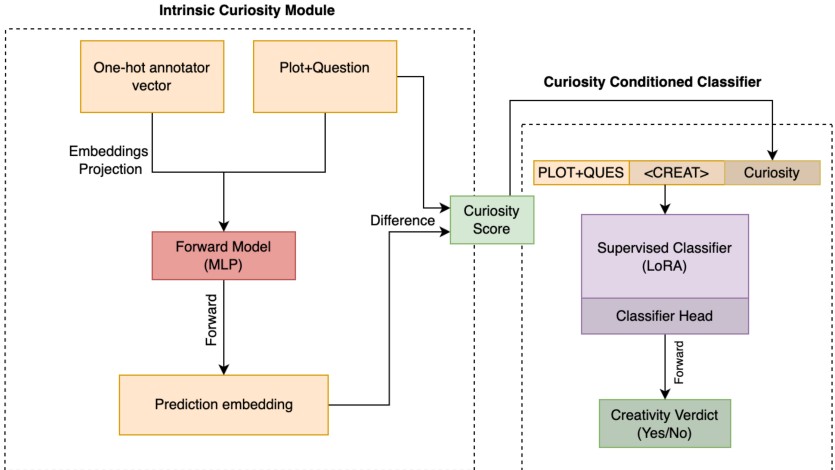

Figure 2: Overview of Architecture during inference for Curiosity Driven LLM-as-a-judge

## 2 METHODOLOGY

In this section, we describe our curiosity-driven LLM-as-a-judge for evaluating creativity in text generation, which combines belief shift estimation with expert attribution. Our method leverages the TTCW dataset Chakrabarty et al. (2024), which is based on the Torrance Test of Creative Thinking

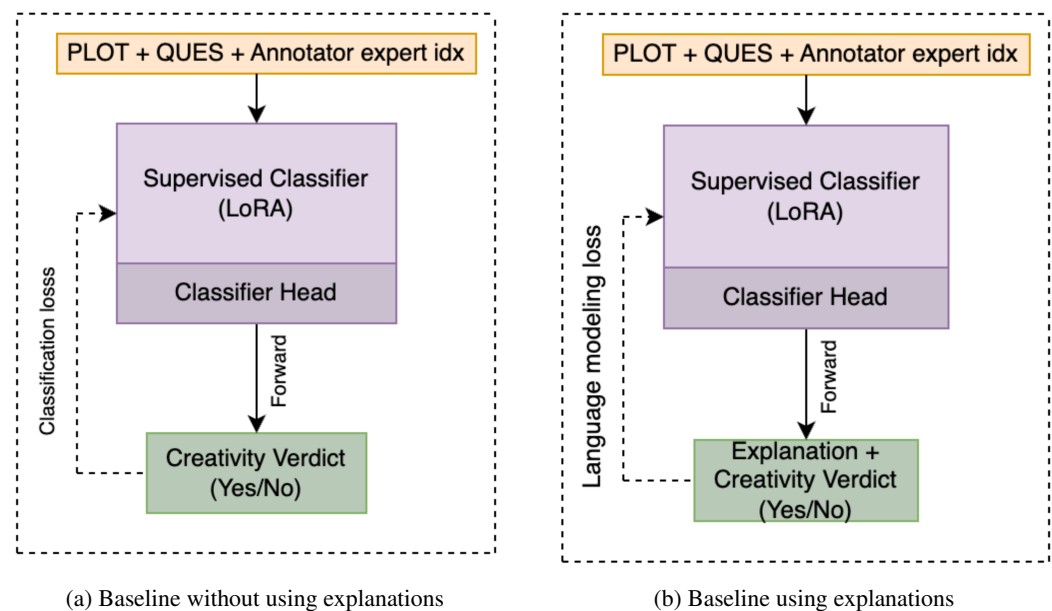

(a) Baseline without using explanations       (b) Baseline using explanations

Figure 3: Comparison of baselines with and without using explanations.

Torrance (1966) but adapted for LLMs. We focus on a subset of five creativity dimensions particularly relevant for evaluating the creative judgments of generative language models. We detail the dataset structure, model architecture, loss functions, and the formulation of our curiosity signal.

## 2.1 DATASET

The TTCW dataset[1] provides expert human-annotated creativity judgments across 14 distinct dimensions. All the distinct dimensions in the TTCW dataset are mentioned in Appendix A.1. For this study, we focus on five dimensions, 3 of which are categorised under Originality and 2 under flexibility: *Originality in Thought*, *Originality in Form*, *Originality in Theme and Content*, *Structural Flexibility*, and *Perspective and Voice Flexibility*. Our analysis is restricted to these five dimensions, encompassing all dimensions under *Originality* and two representative dimensions from *Flexibility*. We picked these 5 dimensions among the 14(Table 4) as these are more subjective in nature and hence the most ideal to evaluate our methodology. We defer exploration of the remaining dimensions to future work. Questions associated with each dimension can be found in appendix 6.

## 2.2 DATA FORMAT AND TASK SETUP

Each example in the dataset consists of a story $S$, a creativity-focused question $Q_d$ specific to dimension $d$, an expert ID $z_i$ where $i \in \{1, 2, 3\}$ for each annotation by an expert, three expert-provided explanations $\mathcal{E} = \{e_1, e_2, e_3\}$, and corresponding binary verdicts $V_i \in \{\text{yes}, \text{no}\}$ for each explanation.

The task is to improve the model's performance on producing judgments similar to that of a particular expert when the model is presented with the story and the creative question

## 2.3 INTRINSIC CURIOSITY MODEL OVERVIEW

Our model operates in two stages:

1. **Belief Shift Estimation (Forward Score)**: The model measures the impact of an expert explanation on their prediction of creativity.

---

[1]Huggingface TTCW dataset

2. **Expert Attribution (Backward Score)**: The model identifies which expert wrote a given explanation.

### 2.3.1 FORWARD SCORE: BELIEF SHIFT VIA COSINE LOSS

We define two states:

- **State A**: Input consisting of the story and question and one-hot vector of the expert ID $z_i$ represented as $(S, Q_d, onehot(z_i))$ where $i \in \{1, 2, 3\}$ as each story-question pair is annotated by 3 experts.

- **State B**: Input augmented with one expert explanation $(S, Q_d, e_i)$ where $i \in \{1, 2, 3\}$.

Let $f_\theta^{(A)} = f_\theta(S, Q_d, onehot(z_i))$ and $f_\theta^{(B)} = f_\theta(S, Q_d, e_i)$, where $f_\theta$ denote the judge's scoring function (logit head) with parameters $\theta$ that maps the input to a scalar judgment logit.

The forward loss is defined as the cosine loss between these two predictions:

$$\mathcal{L}_{\text{forward}} = 1 - \frac{f_\theta^{(A)} \cdot f_\theta^{(B)}}{\|f_\theta^{(A)}\|\|f_\theta^{(B)}\|}$$

This loss captures how much the model's belief about creativity of the story shifts when it incorporates the explanation by the annotator, which we define as the intrinsic curiosity measure.

### 2.3.2 BACKWARD SCORE: EXPERT ATTRIBUTION VIA CROSS-ENTROPY

To help the model to understand the distinct reasoning styles of different experts, we introduce an auxiliary classification task. Given $(S, Q_d, e_i)$, the model predicts the identity of the expert $z_i \in \{1, 2, 3\}$ who authored explanation $e_i$:

$$p_\phi(z_i \mid S, Q_d, e_i) = \text{softmax}(g_\phi(S, Q_d, e_i))$$

The backward loss is the cross-entropy between the predicted and true expert label:

$$\mathcal{L}_{\text{backward}} = -\log p_\phi(z_i \mid S, Q_d, e_i)$$

### 2.3.3 LOSS FUNCTION OF INTRINSIC CURIOSITY MODEL(ICM)

We define the ICM model's loss as a weighted combination of the forward and backward components:

$$\mathcal{L}_{\text{curiosity}} = \mathcal{L}_{\text{forward}} + \lambda \cdot \mathcal{L}_{\text{backward}}$$

where $\lambda$ is a tunable hyperparameter that balances the two objectives. In our experiments we set $\lambda$ as 1.

### 2.3.4 INCORPORATING THE CURIOSITY SIGNAL TO SFT

To evaluate the utility of the learned curiosity signal, we use it as a conditioning input to a supervised fine-tuning (SFT) model trained to predict expert verdicts. For each instance, we append the scalar curiosity score to the original input using a special delimiter token `<CREAT>`, resulting in the following input format:

$$\texttt{Input:} \quad Q_d + S + \texttt{<CREAT>} + Curiosity_{\text{Score}} \quad \longrightarrow \quad \texttt{Target:} \quad V_i$$

$$\text{Curiosity}_{\text{score}} = f_\theta(S, Q_d, e_i) - f_\theta(S, Q_d, \text{onehot(expert\_idx)})$$

$V_i \in \{\texttt{yes}, \texttt{no}\}$ is the binary verdict associated with explanation $e_i$. The model uses the $Curiosity_{\text{Score}}$ as a signal to predict the verdict of the given annotator. We use cross-entropy loss for training this classifier model

## 2.4 INFERENCE

During inference(see Fig 2), the story and creativity-focused questions are first passed through the intrinsic curiosity model (ICM) to compute a curiosity score. This score reflects the model's internal belief shift in response to the input for that particular annotator. The resulting curiosity score is then appended to the original input, using a special delimiter token <CREAT>—and passed to the SFT classifier model. This classifier then predicts the binary creativity verdict (yes or no) for the given story-question pair. .

## 2.5 BASELINE WITH EXPLANATIONS

For the baseline comparison , we use a standard SFT model that produces the explanation and binary verdict given the input(see fig. 3b). The model input is structured as:

$$\texttt{Input:} \quad Q_d + S + z_i \quad \longrightarrow \quad \texttt{Target:} \{V_i, e_i\}$$

At inference time, we provide $Q_d$, $S$, and $z_i$ as input, and the model outputs a JSON structure, from which the predicted verdict is parsed and compared to the ground truth. This baseline is trained using language modeling loss.

## 2.6 BASELINE WITHOUT EXPLANATIONS

We ensure to compare our method against the baseline SFT in a classification setting rather than a causal language model setting to ensure fairness in comparison(see fig. 3a). Since we set up the baseline SFT in a classification setting, we do not include the explanations as neither part of the input or the output of the classification task. In this classification setting we use the question and the story as part of input and the verdict as part of the output.

$$\texttt{Input:} \quad Q_d + S + z_i \quad \longrightarrow \quad \texttt{Target:} \{V_i\}$$

## 2.7 EVALUATION

Evaluating subjective tasks like creativity presents unique challenges, as even human annotators often disagree on what constitutes a "correct" judgment. Rather than attempting to define a universal metric for creativity, our approach embraces this subjectivity by focusing on personalization. We aim to adapt evaluation signals to individual experts by learning from a small number of their labeled examples. This allows us to model subjective preferences more faithfully and use this personalized model to assess creativity in a user-aligned manner. To quantify model performance in capturing individual judgments, we report **Pearson Correlation** Benesty et al. (2009) and **Cohen's** $\kappa$ Cohen (1960), along with **Precision**, **Recall**, and **F1-score**. These metrics enable us to assess both the predictive accuracy and ranking consistency of our models in aligning with subjective human evaluations.

## 3 THEORY: WHY CURIOSITY BEATS USING EXPLANATION TEXT DIRECTLY

Let $e$ denote the expert's explanation, x = $Q_d + S$, $s_{\text{base}}(x) = f_\theta(S, Q_d, \text{onehot}(z_i))$ the pre-explanation logit, and $s_{\text{expl}}(x, e_i) = f_\theta(S, Q_d, e_i)$ the post-explanation logit produced by the model when conditioned on $e$. The $Curiosity_{\text{Score}}$ is defined as the belief shift.

$$\text{Curiosity}_{\text{score}} = f_\theta(S, Q_d, e_i) - f_\theta(S, Q_d, \text{onehot}(z_i)),$$

and *discard* $e$ thereafter. We train a predictor $\hat{p}_\theta(V{=}1 \mid x, \text{Curiosity}_{\text{score}}) = \sigma(h_\theta(x, \text{Curiosity}_{\text{score}}))$ where $V$ is the verdict, $h$ is the LLM judge model and $\sigma$ represents softmax. This yields three advantages grounded in standard theory.

Table 1: ICM method results against the SFT baseline with explanations

| Model | Exp. | LoRA $\alpha$/R | Pearson | Cohen's $\kappa$ | F1 | Precision | Recall |
|---|---|---|---|---|---|---|---|
| Qwen0.5B | SFT | 256/256 | 0.170 ±0.049 | 0.155 ±0.046 | 0.382 ±0.049 | 0.452 ±0.059 | 0.334 ±0.060 |
| | ICM | 32/16 | **0.524** ±0.092 | **0.383** ±0.076 | **0.616** ±0.048 | **0.494** ±0.046 | **0.818** ±0.067 |
| Qwen1.5B | SFT | 256/256 | 0.170 ±0.048 | 0.155 ±0.048 | 0.402 ±0.049 | 0.432 ±0.020 | 0.383 ±0.083 |
| | ICM | 32/16 | **0.587** ±0.061 | **0.406** ±0.065 | **0.629** ±0.045 | **0.506** ±0.045 | **0.836** ±0.056 |
| Qwen3B | SFT | 256/256 | 0.113 ±0.083 | 0.110 ±0.081 | 0.339 ±0.051 | 0.401 ±0.067 | 0.298 ±0.060 |
| | ICM | 32/16 | **0.540** ±0.057 | **0.356** ±0.081 | **0.598** ±0.054 | **0.481** ±0.050 | **0.794** ±0.070 |
| Qwen7B | SFT | 128/128 | 0.160 ±0.050 | 0.168 ±0.085 | 0.371 ±0.021 | 0.443 ±0.050 | 0.324 ±0.038 |
| | ICM | 32/16 | **0.605** ±0.083 | **0.429** ±0.082 | **0.643** ±0.053 | **0.518** ±0.051 | **0.850** ±0.072 |

**(1) Weight-of-evidence sufficiency.** In logit/Bayesian updates, additional information acts *additively* on log-odds via a log-likelihood ratio (*weight of evidence*) (Agresti, 2013):

$$\log \frac{\Pr(V = 1 \mid x, e_i)}{\Pr(V = 0 \mid x, e_i)} \;=\; \log \frac{\Pr(V = 1 \mid x)}{\Pr(V = 0 \mid x)} \;+\; \underbrace{\log \frac{p(e \mid V = 1, x)}{p(e \mid V = 0, x)}}_{\text{weight of evidence}}.$$

In our methodology, $\text{Curiosity}_{\text{Score}} = s_{\text{expl}} - s_{\text{base}}$ is an *empirical estimate* of this increment on the log-odds scale, so it preserves the decision-relevant effect of $e$ while removing lexical/style nuisance. Consequently, conditioning on $\text{Curiosity}_{\text{Score}}$ approximates the theoretically "right" sufficient update in a logistic decision rule (Agresti, 2013).

**(2) Variance reduction via a control-variate effect.** Let $Z$ be the random quantity we wish to estimate more stably (e.g., per-example loss), and let $C = \text{Curiosity}_{\text{Score}}$ be the control signal. With Pearson correlation

$$\rho \;=\; \text{Corr}(Z, C) \;=\; \frac{\text{Cov}(Z, C)}{\sqrt{\text{Var}(Z)\,\text{Var}(C)}} \in [-1, 1],$$

the classic control-variate construction implies that the optimally adjusted estimator $Z^\star = Z - \alpha^\star(C - \mathbb{E}[C])$ achieves

$$\text{Var}(Z^\star) \;=\; \text{Var}(Z)\left(1 - \rho^2\right) \quad \text{at} \quad \alpha^\star = \frac{\text{Cov}(Z, C)}{\text{Var}(C)}.$$

Thus any nonzero correlation with $c$ strictly reduces variance (Owen, 2013, Ch. 8). Here, $Z = \ell_i(\theta)$ (per-example cross-entropy loss) to reduce risk variance. Lower variance improves sample efficiency and stabilizes training.

**(3)Curiosity as a Model of Annotator Behaviour and Generalization** Subjective labels reflect both item difficulty and rater idiosyncrasy. A classic way to formalize this is a random–effects logit (Dawid and Skene, 1979; Agresti, 2013):

$$\text{logit}\,\Pr(V{=}1 \mid x, z_i) \;=\; f(x) \;+\; b_{z_i}(x), \tag{1}$$

where $f(x)$ captures item evidence and $b_a(x)$ represents the (possibly context-dependent) strictness/leniency of annotator $a$. Since the curiosity score is able to model the annotator behaviour without considering the idiosyncrasies of the explanation text, it is able to better generalize to out-of-distribution dimensions for that annotator.

## 4 EXPERIMENTS

We evaluate our Intrinsic Curiosity Modeling (ICM) approach against a supervised fine-tuning (SFT) baseline (see Section 2) across multiple model sizes. For a fair comparison in terms of identical input and outputs, we compare the ICM setup against SFT baseline with explanations. We also compare the ICM setup against FT baseline without explanations in order to ensure the same classification loss is used.

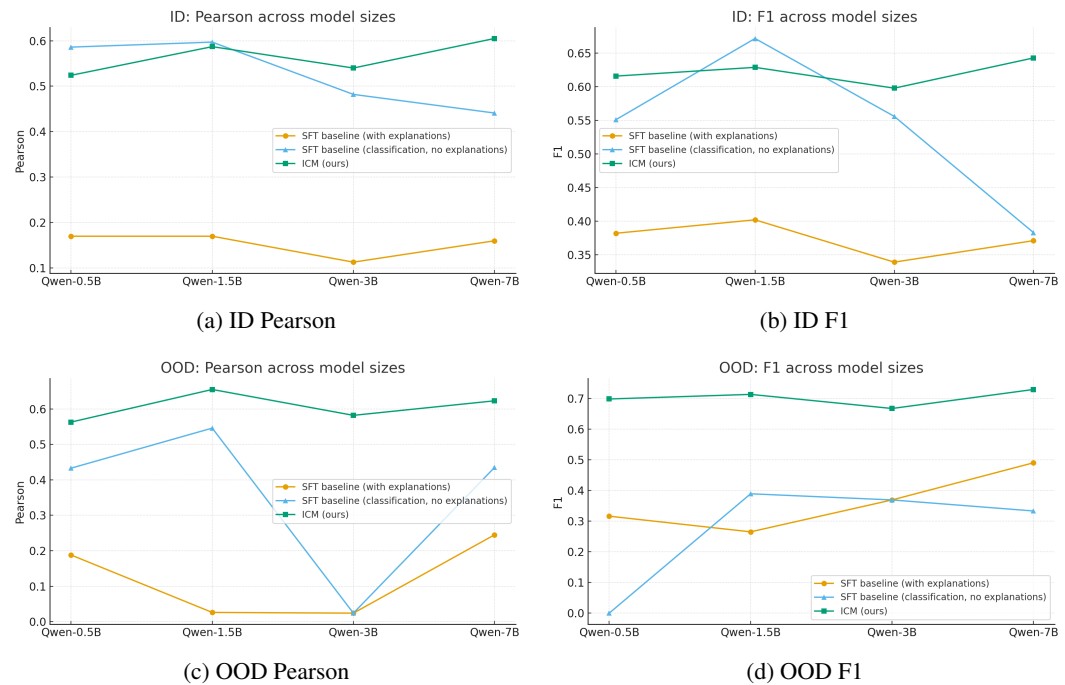

Figure 4: Three-way comparison across model sizes for **ICM (ours)**, **SFT baseline (classification, no explanations)**, and **SFT baseline (with explanations)**. Panels show Pearson and F1 for in-distribution (top) and out-of-distribution (bottom). For exact results of the ID and OOD experiments of *baseline without explanation(classification)*, refer to Table 13 and Table 14

Table 2: Comparison of ICM method against GPT-5 one-shot

| Model | Exp. | Pearson | F1 | Precision | Recall |
|---|---|---|---|---|---|
| Qwen0.5B | ICM | $0.524$ $_{\pm 0.092}$ | $0.616$ $_{\pm 0.048}$ | $0.494$ $_{\pm 0.046}$ | $0.818$ $_{\pm 0.067}$ |
| Qwen1.5B | ICM | $0.587$ $_{\pm 0.061}$ | $0.629$ $_{\pm 0.045}$ | $0.506$ $_{\pm 0.045}$ | $0.836$ $_{\pm 0.056}$ |
| Qwen3B | ICM | $0.540$ $_{\pm 0.057}$ | $0.598$ $_{\pm 0.054}$ | $0.481$ $_{\pm 0.050}$ | $0.794$ $_{\pm 0.070}$ |
| Qwen7B | ICM | $0.605$ $_{\pm 0.083}$ | $0.643$ $_{\pm 0.053}$ | $0.518$ $_{\pm 0.051}$ | $0.850$ $_{\pm 0.072}$ |
| GPT-5 | ICM | $0.2409$ $_{\pm 0.1379}$ | $0.3467$ $_{\pm 0.1592}$ | $0.5698$ $_{\pm 0.2305}$ | $0.2608$ $_{\pm 0.1378}$ |

**Dataset** TTCW contains 48 stories annotated on 5 dimensions with three expert judgments per story–dimension pair, yielding 720 examples. We use 5-fold cross-validation with an 80/20 split, giving approximately 576 training and 144 test items per fold. Because individual folds are small, we report means across folds for all metrics (Table 1; see also Section 2.7). Splits are stratified to preserve the positive/negative label ratio.

**Training setup.** The *baseline with explanations* uses a causal language modeling objective and our ICM model uses a classification objective. We align shared hyperparameters—learning rate, LoRA Hu et al. (2022) rank, and batch size—wherever applicable to ensure comparability. The ICM combined loss uses $\lambda = 1$. All fine-tuning (ICM and SFT baselines) uses LoRA; full details are in Table 5. For the *baseline without explanations*, which also uses a classification loss, we match all of the ICM hyperparameters.

**Compute and precision.** All runs use a single NVIDIA A100 (80 GB) GPU. Mixed precision with **bfloat16** is enabled when supported. When base models are loaded with 8-bit quantization, matrix multiplies in bitsandbytes execute in FP16 while LoRA heads operate in bfloat16.

Table 3: ICM method results against the SFT baseline with explanations on Out-of-distribution data

| Model | Experiment | LoRA $\alpha$/Rank | Pearson | Cohen's $\kappa$ | F1 | Precision | Recall |
|---|---|---|---|---|---|---|---|
| Qwen0.5B | SFT | 256/256 | 0.188 | 0.147 | 0.316 | 0.632 | 0.211 |
| | ICM | 32/16 | **0.563** | **0.458** | **0.698** | **0.625** | **0.790** |
| Qwen1.5B | SFT | 256/256 | 0.026 | 0.023 | 0.265 | 0.423 | 0.193 |
| | ICM | 32/16 | **0.655** | **0.486** | **0.713** | **0.639** | **0.807** |
| Qwen3B | SFT | 256/256 | 0.024 | 0.024 | 0.369 | 0.413 | 0.333 |
| | ICM | 32/16 | **0.582** | **0.403** | **0.667** | **0.597** | **0.754** |
| Qwen7B | SFT | 128/128 | 0.245 | 0.237 | 0.490 | 0.585 | 0.421 |
| | ICM | 32/16 | **0.623** | **0.514** | **0.729** | **0.653** | **0.825** |

**Convergence and reproducibility.** We train to loss convergence in all runs and fix random seeds for data splits and initialization. Hyperparameters and implementation details appear in Table 5.

## 5 ANALYSIS

### 5.1 EFFECT OF MODEL SCALE

From Fig 4 we can see that our ICM method improves across model sizes whereas the *baseline classification method with no explanation* degrades with increase in model size for both ID and OOD settings. The reason why the *baseline classification method with no explanation* maybe degrading with scale is because this method primarily overfits on the small dataset with larger model sizes. Although the *baseline with explanation* improves with increase in model size, it remains uniformly low compared to the ICM method.

### 5.2 GENERALIZATION

To understand the generalization ability of the baseline and the ICM models, we use the same setup as earlier but train the model in both methods on 4 dimensions - *Originality in Form*, *Originality in Theme and Content*, *Structural Flexibility*, and *Perspective and Voice Flexibility*, and test these trained models on the held out dimension of *Originality in Thought*. In this way there is absolutely no data leakage since the dimension the model is tested on was never seen during the training. From figure 4, we can see that gains of the ICM method over both the baseline methods are much more in the OOD settings rather than ID settings. This suggests the generalizability of our method because we are essentially allowing the model to understand the user behavior before predicting which is much more generalizable as compared to both baseline SFT methods.

### 5.3 COMPARISON WITH GPT-5

Table 2 has the results of the ICM setup against GPT-5. We can see that even Qwen-0.5B model is able to beat GPT-5 model across all evaluation metrics except precision. The GPT-5 model was prompted with the same story, question and annotator index along with one shot example(randomly picked from training set) by the same annotator. GPT-5 model was more biased towards the answer "no" and whenever "yes" was predicted, it was almost always wrong. This further proves the effectiveness of our method.

## 6 CONCLUSION AND FUTURE WORK

We introduced a curiosity-driven LLM-as-a-judge for evaluating creativity in text generation, addressing the limitations of baseline SFT for inherently subjective tasks. Our approach leverages a two-part curiosity signal, capturing belief shifts via model responses to expert explanations and incorporating

expert attribution through a backward prediction task. This signal enhances a SFT setup, leading to stronger alignment with human judgments across multiple creativity dimensions in the TTCW dataset. Experiments show that incorporating curiosity-based modeling consistently improves performance across model scales, surpassing standard SFT baselines in both correlation with human ratings and classification accuracy. Not only does it scale with model size, it also improves the performance in out-of-distribution scenarios, where we test the models on one heldout test dimension by training the models on the other 4 creativity dimension. Future work includes extending the curiosity-driven LLM-as-a-judge to other domains like marketing, evaluating novelty of scientific ideas etc,. We also plan to use the curiosity signal as a reward signal in RL setup to further improve our current results.

## 7 LITERATURE REVIEW

The evaluation of creativity in language models builds upon decades of work in creativity research, where the Torrance Tests of Creative Thinking (TTCT) assess fluency, flexibility, originality, and elaboration Torrance (1966), and the Consensual Assessment Technique (CAT) uses aggregated expert judgments, a reliable but labour-intensive process Patterson et al. (2024). The authors of Chakrabarty et al. (2024) adapted TTCT into the Torrance Tests for Creative Writing (TTCW), designing fourteen binary tests and enlisting creative-writing experts to evaluate 48 stories; their study showed that large language models pass these tests three to ten times less often than human writers Chakrabarty et al. (2024), highlighting a sizable gap in creative competence. Alternative evaluation paradigms, such as the Leap-of-Thought (LoT) framework for humorous, associative reasoning, argue that step-by-step chain-of-thought prompting can limit creativity and instead encourage non-sequential "leaps" Zhong et al. (2024). Efforts to automate creativity scoring (e.g., distributional-semantics proxies for novelty) often align weakly with expert judgments, reinforcing the need for human-aligned signals.

Because creativity judgments are *subjective*, collapsing rater perspectives via majority vote can erase systematic, meaningful disagreement. Following work on multi-annotator modeling, we treat annotators as distributions to be modeled rather than aggregated away Mostafazadeh Davani et al. (2022), rather than use the classical aggregation methods that infer a single latent "truth" Whitehill et al. (2009); Hovy et al. (2013). In parallel, recent results caution against naïve *LLM-as-judge* usage: evaluators can recognize and prefer their own generations, introducing self-preference bias Panickssery et al. (2024b). Calibrated autoraters offer a partial mitigation via broad multi-task training and bias auditing Vu et al. (2024). These findings motivate rater-aware or human-anchored evaluation signals for creativity.

Intrinsic-motivation signals from reinforcement learning offer a principled lens on novelty seeking. Information-gain and prediction-error formulations—VIME Houthooft et al. (2017), ICM Pathak et al. (2017), and Random Network Distillation Burda et al. (2018)—are effective for exploration under sparse extrinsic reward. By analogy, curiosity-style signals can inform language evaluation by rewarding "useful novelty" (divergent yet coherent), complementing semantic-distance and rater-based methods. Our work instantiates this by modeling belief shifts when a language model incorporates expert explanations (a prediction-error–like signal) and combining it with expert attribution, yielding a more interpretable and *personalized* measure of creativity.

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

# A APPENDIX

## A.1 DIMENSIONS IN DATASET

In Table 4, all the dimensions that are part of the TTCW dataset are mentioned.

Table 4: Dimensions of TTCW dataset

| Dimension | Facets |
| --- | --- |
| Fluency | Understandability & Coherence |
| | Narrative Pacing |
| | Scene vs Exposition |
| | Literary Devices & Language Proficiency |
| | Narrative Ending |
| Flexibility | Emotional Flexibility |
| | Perspective & Voice Flexibility |
| | Structural Flexibility |
| Originality | Originality in Form |
| | Originality in Thought |
| | Originality in Theme & Content |
| Elaboration | World Building & Setting |
| | Character Development |
| | Rhetorical Complexity |

## A.2 MORE EXPERIMENT AND COMPUTE DETAILS

## A.3 LIMITATIONS

Our study has some limitations that we hope to address in future work. First, the empirical scope is narrow: we evaluate only on TTCW dataset. Our current method is text-only; extending to richer modalities and subjective tasks beyond TTCW remains future work. In addition, the dataset is small (48 stories × 5 dimensions with three expert judgments per story–dimension, totaling 720 instances). We therefore rely on 5-fold cross-validation and report means and deviation across 5 folds.

Table 5: Core hyperparameters used in all runs.

| | |
|---|---|
| max_length | 4096 |
| lora_dropout | 0.1 |
| target_modules | ["q_proj","k_proj","v_proj","o_proj", "gate_proj","up_proj","down_proj"] |
| lr_scheduler | cosine (warmup_ratio $= 0.1$) |
| per_device_train_batch_size | 4 |
| gradient_accumulation_steps | 8 |
| weight_decay | 0.01 |
| max_grad_norm | 0.5 |
| num_train_epochs | 3 |
| seed | 42 |

Finally, model coverage is limited to one family (Qwen2.5 0.5B–7B), leaving generalization across architectures untested, which we aim to do in future work.

### A.4 QUESTION FOR EACH DIMENSION

Table 6: Creativity evaluation categories and questions

| Category | Question |
|---|---|
| Originality in Thought | Is the story an original piece of writing without any cliches? |
| Originality in Form and Structure | Does the story show originality in its form and/or structure? |
| Originality in Theme and Content | Will an average reader of this story obtain a unique and original idea from reading it? |
| Perspective and Voice Flexibility | Does the story provide diverse perspectives, and if there are unlikeable characters, are their perspectives presented convincingly and accurately? |
| Structural Flexibility | Does the story contain turns that are both surprising and appropriate? |

### A.5 STATISTICAL SIGNIFICANCE TESTING

Table 7: Statistical significance test across 5 folds for Qwen-0.5b model

| Metric | SFT(with expl) (mean±SD) | ICM (mean±SD) | $\Delta$ (ICM−SFT) | $p$ (paired $t$) | Statistically significant? |
|---|---|---|---|---|---|
| Pearson | $0.160 \pm 0.055$ | $0.524 \pm 0.092$ | 0.364 | 0.002 | Yes |
| Spearman | $0.160 \pm 0.055$ | $0.484 \pm 0.078$ | 0.324 | $< 0.001$ | Yes |
| F1 | $0.371 \pm 0.054$ | $0.616 \pm 0.048$ | 0.245 | $< 0.001$ | Yes |

### A.6 ICM RESULTS AGAINST SFT BASELINE WITHOUT EXPLANATIONS

### A.7 CURIOSITY SCORES BASED ON NON-FINETUNED BASE QWEN-0.5B MODEL'S PREDICTION AND GROUND TRUTH MATCH AND MISMATCH

### A.8 WHY IS INVERSE MODEL NECESSARY?

When we ablated for the inverse model in our ICM setup with the given expert annotated data we do not see any difference in the results with using the inverse model or without using it. But the inverse model becomes necessary when we have a non-expert annotator like GPT-2, since it helps to clearly distinguish such outliers. This shows that our forward model of the ICM is good enough to distinguish between multiple expert annotators but we do need the inverse model for outlier cases. The details of our experiments can be found in Table 15, we used Qwen-0.5B model for this experiment.

Table 8: Statistical significance test across 5 folds for Qwen-1.5b model

| Metric | SFT(with expl) (mean±SD) | ICM (mean±SD) | $\Delta$ (ICM−SFT) | $p$ (paired $t$) | Statistically significant? |
|---|---|---|---|---|---|
| Pearson | $0.170 \pm 0.058$ | $0.586 \pm 0.064$ | 0.416 | $< 0.001$ | Yes |
| Spearman | $0.170 \pm 0.058$ | $0.522 \pm 0.069$ | 0.352 | $< 0.001$ | Yes |
| F1 | $0.402 \pm 0.050$ | $0.629 \pm 0.045$ | 0.227 | $< 0.001$ | Yes |

Table 9: Statistical significance test across 5 folds for Qwen-3b model.

| Metric | SFT(with expl) (mean±SD) | ICM (mean±SD) | $\Delta$ (ICM−SFT) | $p$ (paired $t$) | Statistically significant? |
|---|---|---|---|---|---|
| Pearson | $0.113 \pm 0.092$ | $0.540 \pm 0.074$ | 0.427 | $< 0.001$ | Yes |
| Spearman | $0.113 \pm 0.092$ | $0.494 \pm 0.091$ | 0.381 | $< 0.001$ | Yes |
| F1 | $0.339 \pm 0.053$ | $0.618 \pm 0.061$ | 0.279 | $< 0.001$ | Yes |

Table 10: Statistical significance test across 5 folds for Qwen-7b model.

| Metric | SFT(with expl) (mean±SD) | ICM (mean±SD) | $\Delta$ (ICM−SFT) | $p$ (paired $t$) | Statistically significant? |
|---|---|---|---|---|---|
| Pearson | $0.170 \pm 0.058$ | $0.606 \pm 0.084$ | 0.436 | $< 0.001$ | Yes |
| Spearman | $0.170 \pm 0.058$ | $0.542 \pm 0.089$ | 0.373 | $< 0.001$ | Yes |
| F1 | $0.381 \pm 0.029$ | $0.663 \pm 0.058$ | 0.282 | $< 0.001$ | Yes |

Table 11: Average passing rate (%) on individual TTCW, based on annotations of 10 creative writing experts across 48 stories; last column reports Fleiss' $\kappa$ (expert agreement).

| Dimension | Test | GPT-3.5 | GPT-4 | Claude v1.3 | New Yorker | Expert $\kappa$ |
|---|---|---|---|---|---|---|
| Fluency | Understandability & Coherence | 22.2 | 33.3 | 55.6 | 91.7 | 0.27 |
| | Narrative Pacing | 8.3 | 52.8 | 61.1 | 94.4 | 0.39 |
| | Scene vs Exposition | 8.3 | 50.0 | 58.3 | 91.7 | 0.27 |
| | Literary Devices & Language | 5.6 | 36.1 | 13.9 | 88.9 | 0.37 |
| | Narrative Ending | 8.3 | 19.4 | 33.3 | 91.7 | 0.48 |
| Flexibility | Emotional Flexibility | 16.7 | 19.4 | 36.1 | 91.7 | 0.32 |
| | Perspective & Voice Flexibility | 8.3 | 16.7 | 19.4 | 72.2 | 0.44 |
| | Structural Flexibility | 11.1 | 19.4 | 30.6 | 88.9 | 0.39 |
| Originality | Originality in Form | 2.8 | 8.3 | 0.0 | 63.9 | 0.41 |
| | Originality in Thought | 2.8 | 44.4 | 19.4 | 91.7 | 0.40 |
| | Originality in Theme & Content | 0.0 | 19.4 | 11.1 | 75.0 | 0.66 |
| Elaboration | World Building & Setting | 16.7 | 41.7 | 58.3 | 94.4 | 0.33 |
| | Character Development | 8.3 | 16.7 | 16.7 | 61.1 | 0.31 |
| | Rhetorical Complexity | 2.8 | 11.1 | 5.6 | 88.9 | 0.66 |
| **Average** | | **8.7** | **27.9** | **30.0** | **84.7** | **0.41** |

Table 12: Correlation between LLM-administered TTCW and expert annotations (Cohen's $\kappa$) on all 48 stories.

| Dimension | Test | GPT-3.5 | GPT-4 | Claude |
|---|---|---|---|---|
| Fluency | Understandability & Coherence | -0.01 | -0.01 | -0.17 |
| | Narrative Pacing | 0.05 | 0.00 | -0.22 |
| | Scene vs Exposition | -0.03 | -0.08 | -0.23 |
| | Literary Devices & Language | 0.04 | -0.09 | -0.11 |
| | Narrative Ending | -0.02 | 0.02 | 0.02 |
| Flexibility | Emotional Flexibility | -0.04 | 0.00 | 0.09 |
| | Perspective & Voice | 0.00 | 0.26 | 0.14 |
| | Structural Flexibility | -0.04 | 0.00 | -0.07 |
| Originality | Originality in Form | 0.08 | 0.09 | 0.03 |
| | Originality in Thought | 0.19 | 0.31 | 0.15 |
| | Originality in Theme & Content | 0.06 | -0.01 | 0.18 |
| Elaboration | World Building & Setting | 0.00 | 0.00 | 0.09 |
| | Character Development | -0.08 | 0.02 | 0.00 |
| | Rhetorical Complexity | 0.00 | 0.00 | 0.02 |
| **Average** | | **0.016** | **0.035** | **-0.006** |

Table 13: ICM method results against the SFT baseline without explanations (classification). Means±SD are shown where SD was available from 5-fold runs.

| Model | Experiment type | pearson | precision | recall | f1 |
|---|---|---|---|---|---|
| Qwen-0.5B (SFT-Classification) | ID | **0.586 ± 0.085** | **0.769** | 0.461 | 0.551 ± 0.198 |
| Qwen-0.5B (ICM) | ID | 0.524 ± 0.092 | 0.494 | **0.818** | **0.616 ± 0.048** |
| Qwen-1.5B (SFT-Classification) | ID | **0.602 ± 0.064** | **0.787** | 0.602 | **0.663 ± 0.070** |
| Qwen-1.5B (ICM) | ID | 0.586 ± 0.064 | 0.481 | **0.794** | 0.629 ± 0.045 |
| Qwen-3B (SFT-Classification) | ID | 0.482 ± 0.160 | **0.670** | 0.573 | 0.556 ± 0.094 |
| Qwen-3B (ICM) | ID | **0.540 ± 0.074** | 0.481 | **0.794** | **0.618 ± 0.061** |
| Qwen-7B (SFT-Classification) | ID | 0.441 ± 0.130 | **0.535** | 0.342 | 0.383 ± 0.251 |
| Qwen-7B (ICM) | ID | **0.606 ± 0.084** | 0.518 | **0.850** | **0.663 ± 0.058** |

**Note.** SDs for *precision* and *recall* were not available in the provided per-fold summaries; once those per-fold values are supplied, I will fill in their ± SD as well. Pearson/F1 SDs are computed across 5 folds.

Table 14: ICM method results against the SFT baseline without explanations(classification) on Out-of-distribution data

| Model | Experiment type | pearson | precision | recall | f1 |
|---|---|---|---|---|---|
| Qwen-0.5B(SFT-Classifcation) | OOD | 0.433 | 0.000 | 0.000 | 0.000 |
| Qwen-0.5B(ICM) | OOD | **0.563** | **0.625** | **0.790** | **0.698** |
| Qwen-1.5B(SFT-Classifcation) | OOD | 0.604 | **0.962** | 0.439 | 0.602 |
| Qwen-1.5B(ICM) | OOD | **0.655** | 0.639 | **0.807** | **0.713** |
| Qwen-3B(SFT-Classifcation) | OOD | 0.546 | **0.933** | 0.246 | 0.389 |
| Qwen-3B(ICM) | OOD | **0.582** | 0.597 | **0.754** | **0.667** |
| Qwen-7B(SFT-Classifcation) | OOD | 0.435 | 0.800 | 0.211 | 0.333 |
| Qwen-7B(ICM) | OOD | **0.623** | **0.653** | **0.825** | **0.729** |

Table 15: Inverse model ablations

| Method | Annotations | Pearson | Precision | Recall | F1 | Cohen's $\kappa$ |
|---|---|---|---|---|---|---|
| ICM with Inverse | Without GPT-2 | 0.503 ± 0.014 | 0.552 ± 0.014 | 0.728 ± 0.017 | 0.628 ± 0.015 | 0.347 ± 0.027 |
| ICM without Inverse | Without GPT-2 | 0.500 ± 0.027 | 0.551 ± 0.011 | 0.727 ± 0.009 | 0.627 ± 0.010 | 0.346 ± 0.017 |
| ICM with Inverse | With GPT-2 | 0.151 ± 0.300 | 0.153 ± 0.265 | 0.233 ± 0.403 | 0.185 ± 0.320 | 0.093 ± 0.166 |
| ICM without Inverse | With GPT-2 | 0.002 ± 0.041 | 0.333 ± 0.577 | 0.001 ± 0.002 | 0.002 ± 0.004 | 0.000 ± 0.004 |

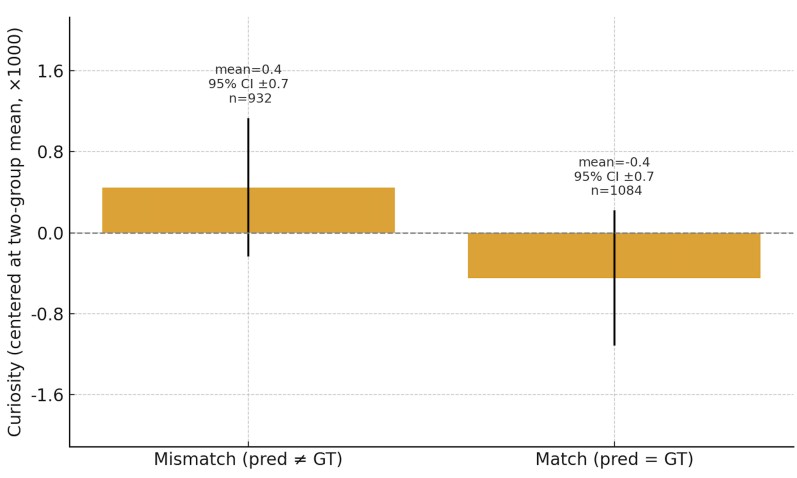

Figure 5: Curiosity scores based on match and mismatch of predictions from Qwen-0.5B base non-finetuned model and the ground truth

