# OpenReview forum: "Curiosity-Driven LLM-as-a-judge for Personalized Creative Judgment"
_ICLR.cc/2026/Conference — ICLR 2026 Conference Withdrawn Submission_

### Official Review · Reviewer_V3Dr · 2025-10-27

**Soundness:** 2
**Presentation:** 3
**Contribution:** 2
**Rating:** 4
**Confidence:** 4

**Summary:**

The work introduces a method leveraging a "curiosity driven" method to evaluate creative fiction along specific dimensions, when compared to expert judgements. The curiosity driven methodology relies on expert identities and the explanations they've written accompanying their binary judgements. The method leverages these signals to train expert-specific predictors. The method is demonstrated to outperform baselines that do not leverage such signals during training, such as standard SFT.

**Strengths:**

- The study of creative writing evaluation is important, and the work introduces the idea of using expert rationales which is an interesting and underexplored direction.
- The magnitude of the results are compelling as the method significantly outperforms the baselines including a larger and costlier LLM (GPT5), though come with questions (see weaknesses and questions).

**Weaknesses:**

- The model describes the TTCW dataset as having 3 experts. However, the paper that introduced the TTCW dataset (https://arxiv.org/pdf/2309.14556) describes that 10 experts participated in the annotation, with any of 3 experts assigned to any given story. Therefore I believe there might be a methodological flaw in the paper, as there should really be 10 expert embeddings learned, and not 3. Are the authors aware of this and have they compensated for this in some way that is not described? It is likely that the annotator number (1,2,3) is arbitrary.
- The work focuses on 5 of the 14 TTCWs, and the authors argue these were chosen as they are more open-ended, yet one would argue that the method would likely work better on more interpretable tests, and this subselection could be seen as cherry-picking rather than a justified choice. Have the authors attempted the methodology on the other tests (in the Fluency, Flexibility categories). Does the method generalize? If not, is it a cost concern or something else?

**Questions:**

- See the questions listed in the Weaknesses section.
- Currently, the explanations from experts are only used during training. Have you considered: the ability to generate explanations according to an expert, or if explanations were somehow available at test time, how that would impact performance?

---

### Official Review · Reviewer_1YV9 · 2025-10-27

**Soundness:** 3
**Presentation:** 2
**Contribution:** 3
**Rating:** 4
**Confidence:** 2

**Summary:**

This paper proposes a curiosity-driven framework for improving the ability of LLMs to make personalized creative judgments. By modeling the belief shift that occurs when an expert explanation “surprises” the model, it introduces a CuriosityScore that captures creativity in human evaluation.

**Strengths:**

1. The paper uses “curiosity” as a computational signal to improve subjective judgment modeling. This integration of cognitive science concepts with modern LLM training is original and intellectually stimulating.
2. The analysis section grounds the CuriosityScore in established statistical principles.

**Weaknesses:**

1. The TTCW dataset is small (48 stories * 3 nnotators) and highly specific to English creative writing. This raises concerns about the robustness and generality of the results. It is unclear whether the findings would hold for other subjective domains.
2. While the paper shows correlations between high curiosity and high creativity, it does not deeply analyze why certain examples trigger high curiosity. A more thorough visualization analysis would help readers understand what “curiosity” actually encodes.
3. LINE 218：No dots. LINE 227: Two dots.

**Questions:**

1. The paper argues that higher model curiosity indicates higher creative novelty. Could this effect simply arise because those samples are rarer or less frequent in the dataset?
2. How sensitive is the method to the quality or style of expert explanations? For example, if explanations are noisy or stylistically inconsistent, does the curiosity signal still generalize well?
3. Could the CuriosityScore be computed without explicit explanations, e.g., by using the model’s internal prediction uncertainty as a proxy?
4. Does the model ever exhibit pathological curiosity—i.e., being overly “surprised” by nonsensical or incoherent inputs? If so, how is this mitigated?

---

### Official Review · Reviewer_RX53 · 2025-10-31

**Soundness:** 1
**Presentation:** 1
**Contribution:** 1
**Rating:** 2
**Confidence:** 3

**Summary:**

Fundamentally, the paper is trying to predict the ratings of expert judges for creative writing passages along dimensions in the Torrance Test for Creativity, originality, and flexibility.

The authors propose to do this by introducing an Intrinsic Curiosity Module. This is learned by optimizing two losses: (a) minimizing the 'forward' loss which encourages high cosine similarity between the predictions on the one-hot encoder of author identity and that corresponding author's explanation, (b) minimizing the 'backward' loss which increases the probability of correctly identifying the expert from the explanation. This is then used to create a Curiosity Score that quantifies the prediction error of the 'forward' model at predicting the explanation. This is then appended as a scalar value to the example during SFT to obtain predictions for creativity ratings.

The approach improves performance on the dataset over an SFT baseline and one-shot GPT-5 when evaluated with k-fold cross validation.

The paper is a good faith effort at a hard problem but overall is both missing simple baselines and is hard to follow in large parts (mismatched notation and difficult ordering of sections). See sections below for specfics.

**Strengths:**

1. The paper attacks the well-motivated problem that preferences of expert annotators might not be well modeled by contemporary reward models in subjective tasks. They propose a novel approach that quantifies and uses their definition of curiosity to make predictions.

**Weaknesses:**

1. The paper is hard to follow with difficult notation and terminology that are non-standard and not explained in sufficient detail.

2. The paper does k-fold cross validation, but it does not hold out other annotations for the same example. So you might be seeing a test annotation from another annotator which can cause leakage and unreliable results. Section 5.2 reports results on another label which is held out, but it is unclear how correlated this label is with the other dimensions on the same input examples. The simple way to mitigate this would be to use the entire existing data for training and collect a small expert test set to confirm the results.

3. There are multiple missing, low-hanging, reward model baselines than just one-shot GPT-5. At the very least, we can expect a few-shot baseline, the un-finetuned Qwen performance and potentially even the methods/models evaluated in [1]. I would also be curious of the actual performance of the top-performing models on RewardBench, even if the particular dataset is specialized to expert ratings.

4. A lot of the modifications proposed are overly complicated and not well motivated. Section 3 provides some insight into the authors' thinking but it reads more as post-hoc explanation of intuition than grounding.

5. Given that the task is fundamentally just a {yes, no} prediction, shouldn't the random chance baseline be a lot stronger than what is reported for SFT?

[1] Chakrabarty, Tuhin, Philippe Laban, and Chien-Sheng Wu. "Ai-slop to ai-polish? aligning language models through edit-based writing rewards and test-time computation." arXiv preprint arXiv:2504.07532 (2025).

**Questions:**

1) Formatting of citations (L.28, 31, 35, Section 7) is incorrect.

2) The formulation is confusing in the use of $f_\theta$ on both sides of the equation, given that I think you mean to say that $f_a$ and $f_b$ are obtained from applying $f_\theta$ to the corresponding input, i.e., two separate models that consume different kinds of input.

3) L.182-183 - The loss just encourages higher similarity between the one-hot encoding of the author and the explanation. I don't follow how it connects with "This loss captures how much the model’s belief about creativity of the story shifts when it incorporates the explanation by the annotator, which we define as the intrinsic curiosity measure."

4) The curiosity score in L.213 is only properly defined in L.266 in the section after. And this is compounded because you use the term curiosity measure in L.186 in a different context. Keeping terminology consistent and defined at the right time would significantly improve readability.

5) The ordering of the sections makes it challenging to follow. Section 3 might be better placed in between 2.4 and 2.5 where there is a shift to experimental baselines.

---

### Official Review · Reviewer_553v · 2025-11-04

**Soundness:** 2
**Presentation:** 2
**Contribution:** 2
**Rating:** 4
**Confidence:** 5

**Summary:**

The authors propose a novel curiosity-driven LLM-as-a-judge for evaluating creative writing which is personlized to each individual's creative judgments. They use the Torrance Test of Creative Thinking(TTCW) benchmark which has stories annotated by expert humans across various subjective dimensions , to test their hypothesis. They show that their method enables models across various sizes, to learn the nuanced creative judgments of different individuals, by showing improvements over baseline supervised finetuning(SFT) method across various evaluation metrics like Pearson correlation, Cohen's and F1 values. Authors claim their method is especially useful in subjective evaluations where not all the annotators agree with each other.

**Strengths:**

The paper addresses an important gap in subjective evaluation where authors recognize that creativity judgments are inherently personal and shouldn't be collapsed into a single "ground truth." I like this approach of pluralism

**Weaknesses:**

While ambitious the experimental scale is somewhat limited. TTCW was designed for a different purpose. It has 48 stories so any ML approach is likely to overfit to just this data? Why cant you include other benchmarks for long form creative writing ? I think such a small corpus doesnt add the strength of evidence for strong empirical claims

Authors are only experimenting with Qwen models tested (0.5B-7B). For stronger claims n>1 model family is required

Authors state uriosity is the "difference" between embeddings (Section 2.3.4) but trains using cosine loss (Section 2.3.1). This inconsistency is a bit confusing

Section A.8 shows the inverse model (expert attribution) has no effect with real experts but only with synthetic GPT-2 annotations. This means the backward loss may be unnecessary for the core use case. So I am wondering the motivation

No ablation showing curiosity signal alone vs. expert ID conditioning/ concatenating expert ID to the input without curiosity

While authors claim OOD test is  performed on one held-out dimension from the same 48 stories, this isnt really OOD

**Questions:**

NA

---

### Note · Authors · 2025-11-21

**Comment:**

We thank the reviewers for their careful evaluations and helpful suggestions. Owing to time constraints that prevent us from running the full set of recommended experiments, we are withdrawing this submission and will resubmit after a more complete revision.

**Withdrawal Confirmation:**

I have read and agree with the venue's withdrawal policy on behalf of myself and my co-authors.